# Genome-Wide Characterization of *bHLH* Family Genes and Expression Analysis in Response to Osmotic Stress in *Betula platyphylla*

**DOI:** 10.3390/plants12213687

**Published:** 2023-10-25

**Authors:** Leifei Zhao, Weiyi Bi, Yaqi Jia, Jingjing Shi, Yao Chi, Mingyu Yu, Chao Wang

**Affiliations:** State Key Laboratory of Tree Genetics and Breeding, Northeast Forestry University, Harbin 150040, China; 13029729795@139.com (L.Z.); bi19990216@163.com (W.B.); yaqi921212@163.com (Y.J.); cy159447@163.com (Y.C.); mingyu678@126.com (M.Y.)

**Keywords:** bHLH, birch, drought stress, qRT-PCR, RNA-seq, genome-wide

## Abstract

The bHLH family, as a superfamily of transcription factors (TFs), has special functional characteristics in plants and plays a crucial role in a plant’s growth and development and helping the plant cope with various stresses. In this study, 128 *bHLH* family genes were screened in the birch (*B. platyphylla*) genome using conservative domain scan and blast analysis. These genes are clustered into 21 subfamilies based on the phylogenetic tree construction and are unevenly distributed among the 14 birch chromosomes. In all, 22 segmental duplication pairs with 27 *BpbHLH* genes were identified. The duplications were distributed on eight chromosomes. Analysis of gene structures and protein motifs revealed intra-group conservation of BpbHLHs. Of the *BpbHLH* family genes, 16 contain only one intron each. The *BPChr14G06667* gene contains the most introns, that is, 19. The cis-elements, which respond to plant hormones, light, defense, and stress, were found on the promoter of *BHLH* family genes. As per RNA-seq data analysis, under PEG osmotic stress, most *BpbHLH* genes were differentially expressed, and eight were highly differentially expressed. The qRT-PCR analysis results further indicated that *BPChr06G09475* was the gene with the highest expression level in leaves, roots, and stems, and that the expression of these eight genes was higher in leaves than in roots and stems and upregulated in all three tissues under osmotic stress compared to the controls. The above analysis suggests that the *BpbHLH* family genes have a certain biological effect under drought stress that provides a basis for molecular breeding for stress resistance in birch.

## 1. Introduction

Changes in the environment affect the growth and development of plants [1]. Abiotic stress includes salt stress, drought stress, chilling stress, metal ion stress, and mechanical damage [2,3]. Woody plants have to experience various abiotic stresses due to their immobility and perennial characteristics. However, woody plants have evolved a series of specific regulation pathways in physiological and molecular mechanisms to deal with adverse environments [4]. Transcription factors (TFs), also known as trans-acting factors, refer to DNA-binding proteins that can specifically interact with cis-acting elements of eukaryotic genes and activate or inhibit gene transcription. TFs play a key role in the stress response regulation network and signal pathway in eukaryotic organisms and activate the plant’s response to environmental changes by regulating the expression of resistance-related genes [5].

Basic/helix–loop–helix (bHLH) TFs are widely present in animals and plants and are named after their basic/helix–circular–helix domain. This domain consists of approximately 60 amino acids and includes two different functional areas: BASIC and HLH regions [6,7,8]. bHLH TFs are usually divided into six groups (I to VI) [9]. Members of Group I can bind to the E-box (CAGCTG). Members of Group II can bind to the G-box (CACGTG), and many do not have any function [10,11]. Group III contains TFs with a protein–protein interaction domain that cannot bind to the core sequence (bHLH GTG/NGCGTG) of the E-box. Group IV members contain HLH regions but lack basic domains [12]. Group V consists of the WRPW-bHLH protein. This protein has a front residual in the basic area and binds to the N-box before combining with the E-box (CACGGC/CACGAC). Group VI includes the Collier/OLF1/EBF-bHLH (COE-bHLH) protein, which is very different from the I-V protein and can work in combination with DNA and dualization [13,14,15].

The bHLH family is the second-largest family in plant TFs. Basic spiral cycle spiral proteins have special functional characteristics in many plants and play a vital role in their growth and development and response to various stresses [16]. The *DYT1* gene encodes putative bHLH-TF-regulated *Arabidopsis* tapetum development [17], EMBRYO1 encodes a new basic helix–loop–helix protein that controls embryo growth [18], and *HECATE* genes regulate female reproductive tract development in *Arabidopsis* [19]. *Laxpanicle* (*LAX*) of *Oryza* sativa is a TF for the bHLH families and is the main regulatory factor in controlling top growth in plants. *OSB1* and *OSB2* can participate in biological synthesis and regulate anthocyanin biosynthesis [20]. When plants suffer abiotic stress, *bHLH* genes influence their stress resistance. *PebHLH56* played a role in the response of *Passiflora edulis Sim* to cold stress [21]. *AtICE1* was involved in tolerance to freezing in *Arabidopsis* [22]. The ectopic expression of cryptochrome-interacting bHLH *AcCIB2* in *Arabidopsis* and complementation of the *AtCIB2* mutant verified the involvement of *AcCIB2* in photomorphogenesis and abiotic stress response [23]. Overexpression of AtAIB, a nuclear-localized bHLH protein in *Arabidopsis*, led to increased drought tolerance in *Arabidopsis* [24]. *AtbHLH92* (*AT5g43650*) mutants were more sensitive to mannitol, and these mutants also showed increased electrolyte leakage following NaCl treatments [25]. *AtbHLH17* transgenic lines exhibited enhanced tolerance to mannitol stress, with significantly higher root growth observed in transgenics [26]. A relative expression analysis in buckwheat showed that *FTbHLH3* may be used as a positive regulatory factor for ABA-dependent drought/oxidation tolerance in the genetically modified *Arabidopsis* [27]. The tissue-specific relative expression of *TabHLH49* in wheat responded to drought stress [28]. Research has shown that *OsbHLH148* enhances drought tolerance in rice through the jasmonate signaling pathway, which is composed of OsbHLH148–OsJAZ1–OsCOI1 modules [29]. *BVbHLH93* played a key role in improving salt stress tolerance by enhancing antioxidant enzymes and reducing the generation of oxygen (ROS) [30]. bHLH TFs in plants such as *Tamarix chinensis*, 84K poplar, *Larix olgensis*, and *Chrysanthemum morifolium* respond to abiotic stress reactions to help these plants better adapt to adversity through a series of regulatory mechanisms. These studies have shown that bHLH TFs play important roles in plant response to abiotic stress. However, there are many members in the *bHLH* gene family. So, it is of great significance to identify the members involved in abiotic stress and study the relationship between their gene structure, expression, and gene function to explore the coordinated regulation of family genes and further serve molecular breeding. 

Birch is a rapidly growing deciduous tree distributed in the northern temperate zone. It is characterized by rapid growth, beautiful shapes, and fine wood properties and provides outstanding ecological benefits. It can be used as a raw material for plywood, joinery, furniture, veneer, papermaking, etc. The bark of the tree produces tannin [31] and birch bark oil [32]. Birch bark is commonly used to make household utensils, and birch juice can be used for medicinal purposes, with health functions. Birch is also popular in landscaping. There are a variety of environmental stresses, especially the drought in the northern region of China, and it is increasingly important to improve the drought resistance of wood plants used in afforestation for landscape or timber. Molecular breeding for stress resistance has become one of the important methods for the genetic improvement of woody plants. Breeding birch more adaptable to environmental changes using molecular breeding techniques to anchor key drought resistance genes in birch may also help, but the drought resistance mechanism of white birch is still unclear. High-throughput sequencing technologies and advanced bioinformatics analysis technologies have provided unprecedented opportunities to leverage the important and unique development and differentiation-related issues of forest trees. The large amounts of data will help to study molecular mechanisms [33]. In this study, a total of 128 *bHLH* genes were identified by bioinformatics. Then, the members that respond to osmotic stress were screened. These *BpbHLH* genes are potential resources for genetically improving birch to resist drought.

## 2. Results

### 2.1. Identification of bHLH Genes in Birch 

To identify the *bHLH* family genes in birch, local HMM search with the HMM file (PF00010) was performed against birch genome, the existence of the conserved bHLH domain was confirmed using SMART and NCBI Batch CD-Search tools, and redundant sequences were removed. In all, 128 putative bHLH protein sequences were identified. The predicted BpbHLH proteins ranged from 66 (BpChr11G07048) to 1487 (BPChr14G06667) amino acids, the isoelectric points ranged from 4.67 (BPChr01G22813) to 11.51 (BPChr05G22456), and the molecular weights ranged from 7346.39 Da (BpChr11G07048) to 164407.6 Da (BPChr14G06667) (Appendix A).

### 2.2. Chromosomal Distribution and Collinearity Analysis of BpbHLH Genes

The *BpbHLH* genes were mapped to the corresponding chromosomes, and 128 *BpbHLH* genes were randomly distributed to all 14 birch chromosomes in a scattered manner (Figure 1A). The maximum number of *BpbHLH* genes, i.e., 20, were distributed on chromosome 6, followed by chromosome 8. Chromosome 10 and chromosome 13 had only four genes each. Gene duplication analysis can provide information regarding evolution during the expansion of the *BpbHLH* genes. In this study, the collinearity of *BpbHLH* family genes was analyzed using the Circos tool in the TBtools 1.045 software. In all, 22 segmental duplication pairs with 27 *BpbHLH* genes were identified and duplications distributed on eight chromosomes (Figure 1B). From these results, it could be concluded that some *BpbHLH* genes are produced by gene duplication events, with segmental duplication playing an essential role in the evolution of the *BpbHLH* genes.

### 2.3. Phylogenetic Tree Analysis and Gene Structure Organization of BpbHLH Proteins

To investigate the evolutionary relationship between BpbHLH members, the amino acid sequences of the bHLH family from *Arabidopsis* and the birch genome were used to construct an NJ phylogenetic tree. The phylogenetic tree showed that these bHLH proteins can be clustered into 21 subfamilies, with 1 to 12 proteins per subfamily. The five main subfamilies are 2, 3, 15, 18, and 19. Subfamily 18 contains the maximum number of BpbHLH proteins: 11. There is only one BpbHLH member in subfamily 20 (Appendix A). To further understand the structural composition of the *BpbHLH* genes, the online GSDS tool was used to display their exon and intron structures and construct a structural distribution map (Figure 2) based on the phylogenetic tree (Appendix A). The results showed that the *BpbHLH* gene family has great gene structural diversity and the number of introns also varies a lot, with 16 of the *BpbHLH* family genes containing only 1 intron each and the *BPchr14G06667* gene containing the most introns: 19. The distribution patterns of exons were relatively conserved in group I and VI, and the genes in these groups had a high similarity in terms of exon number, pattern, and length.

### 2.4. Conserve Motifs and Promoter Cis-Elements Analysis

To characterize potential conserved motifs of the BpbHLH family, protein sequences of BpbHLH members were analyzed using the MEME software (http://meme-suite.org/, 13 March 2021) online. A total of eight conserved motifs (numbered 1 to 8) were identified (Figure 3A). The distribution of the important conserved motifs among BpbHLH proteins showed that members within the same group exhibited highly similar motif distributions (Figure 3A). There were 119 bHLH proteins containing the basic motif and 123 proteins containing the helix1 motif, representing highly conserved domains. Of the BpbHLH genes, 9.3% and 39.1% contained motif 3 and the helix2 motif, respectively. These results indicate the good conservation of the bHLH gene family in plant species. In addition, conserved protein motifs may contribute to their similar functions.

The cis-acting elements on the promoters can reflect the potential function and transcriptional regulation mechanism of genes (Figure 3B). In this study, the promoters of *BpbHLH* genes contained a large number of MYB- and MYC-binding sites and cis-elements that respond to plant hormones, such as MeJA, ABA, SA, and GA, suggesting that *BpbHLH* genes might be regulated by MYB TFs and participate in multiple hormone signaling pathways. In addition, several genes contained a large number of light-responsive elements, the case in point being *BPChr01G13692*. Defense and stress response elements also appeared in large numbers in these promoters, for instance, the G-box and the E-box, implying their roles in response to stress.

### 2.5. Relative Expression Analysis of BpbHLH Genes under Osmotic Stress 

To elucidate the potential roles of the *BpbHLH* genes in response to osmotic stress, the RNA-seq data of birch under 20% (*w*/*v*) PEG_6000_ stress for 0, 2, 4, 6, and 9 h were downloaded from Jia et al. [34]. On the basis of differentially expressed gene (DEG) analysis, the relative expression profiles of BpbHLH TFs exposed to 20% (*w*/*v*) PEG_6000_ stress for 2, 4, 6, and 9 h were constructed. The results showed that 121 *bHLH* genes, accounting for 94.5% of the total, responded to osmotic stress and showed five expression patterns (Figure 4). Among them, 11 *BpbHLH* genes showed upregulated expression as the treatment time increased (Class B), while 18 *BpbHLH* genes showed downregulated expression patterns (Class A). Additionally, the expression levels of 56 *BpbHLH* genes showed an initial increase followed by a decrease over time (Class D), and those of 24 *BpbHLH* genes showed an initial decrease followed by an increase over time (Class C). The transcript levels of 12 *BpbHLH* genes did not show any change under osmotic stress over time (Class E). These results suggest that most of the *BpbHLH* genes may be involved in the regulation in response to drought stress and play different roles.

### 2.6. Analysis of Relative Expression Patterns of Eight bHLH Genes Selected in Roots, Stems, and Leaves

To explore the potential roles of *BpbHLH* genes in plant development and lay the foundation for exploring their molecular mechanisms, eight *BpbHLH* genes representing different subfamilies (Figure 2) and expression patterns (Figure 4) were selected on the basis of the phylogenetic tree and the RNA-seq expression profile. The expression levels of *BpChr12G24135* (Group III) and *BpChr06G09470* (Group VI) were low in the transcriptomes. However, *BpChr06G09475* (Group VI) and *BpChr08G11237* (Group VI) had high expression levels at each treatment time. *BpChr01G18056* (Group III), *BpChr11G07048* (Group IV), *BpChr12G25787* (Group IV), and *BpChr08G16126* (Group II), with moderate expression levels, were also selected. Most members of Group I showed extremely low expression levels in the heatmap, and the changes in the expression levels of Group V members were irregular in the heatmap. So, we did not select any *BpbHLH* genes from these two groups. The qRT-PCR was used to detect the expression levels of the eight selected *bHLH* genes in roots, stems, and leaves of birch. The results showed that *BPChr06G09475* was the gene with the highest expression levels in all three tissues, while the expression levels of *BPChr12G24135* were the lowest (Figure 5), consistent with the results of transcriptome analysis. The expression levels of *BPChr01G18056*, *BPChr06G09475*, *BPChr06G09470*, and *BPChr11G07048* were higher in leaves than in other tissues; *BPChr08G16126* and *BPChr12G25787* were mainly expressed in roots; and the expression levels of *BPChr08G11237* and *BPChr12G24135* were mainly higher in the stems (Figure 5). These results show that the expression of *BpbHLH* genes has tissue-specific characteristics. No differences were observed in the patterns of *BpbHLH* genes among groups.

### 2.7. Analysis of Relative Expression Patterns of Eight Selected BpbHLH Genes in Roots, Stems, and Leaves under PEG_6000_ Stress

To screen the key *BpbHLH* genes in response to osmotic stress and explore the tissue response patterns of eight *BpbHLH* genes under PEG_6000_ treatment, the expression patterns of the eight genes in response to osmotic stresses in roots, stems, and leaves were investigated using qRT-PCR (Figure 6). Under PEG_6000_ treatment, all eight genes had the earliest expression in roots. Among them, *BPChr06G09475*, *BPChr08G11237*, and *BPChr01G18056* responded significantly at 6 h of treatment. The eight *BpbHLH* genes responded relatively late in the leaves, reaching their peak expression at 72 h of treatment, and the expression levels were significantly upregulated (the expression can reach tens of thousands of times compared to the controls), suggesting that the eight genes play a crucial role in the osmotic resistance process of birch leaves. *BPChr12G24135*, *BPChr06G09470*, *BPChr11G07048*, and *BPChr12G25787* were the most significantly upregulated genes in all three tissues after treatment, with *BPChr06G09470*, *BPChr11G07048*, and *BPChr12G25787* being the genes with relatively moderate expression levels in the three tissues and *BPChr12G24135* being the genes with relatively lowest expression levels in the three tissues prior to treatment (Figure 5). The results showed the important role of *BPChr06G09470*, *BPChr11G07048*, and *BPChr12G25787* in the development of birch roots, stems, and leaves and their response to osmotic. As the gene with the highest relative expression in the three tissues, *BPChr06G09475* has a lower response to stress in roots and stems than other genes, except in leaves. Its role in the response of birch leaves to drought is worth exploring. As the only one selected from the Group II gene, *BPChr08G16126* was significantly downregulated in roots and stems under stress treatment, indicating its functional specificity.

## 3. Discussion

Drought is the main abiotic stress that restricts plant growth and development. Therefore, identifying key genes for drought resistance in plants is of great significance for improving their drought resistance. The *bHLH* family genes are among the most common TF families in higher plants, and research shows that bHLH TFs are involved in plant drought resistance regulation. Research has shown that *PxbHLH02* from *Populus simonii* × *Populus nigra* functions as a positive regulator of drought stress responses by regulating stomatal aperture and promoting ABA signal transduction [35]. The expression of the apple bHLH TF *MdCIB1* was induced under drought stress, and the ectopic expression of *MdCIB1* in *Arabidopsis* can protect plants from penetrating and oxidative damage under drought stress [36]. The heterogeneous relative expression of *MFbHLH38* in *Myrica rubra*, which is upregulated under stress treatment and improves the tolerance of *Arabidopsis* for drought stress and salt coercion [37]. In addition, the increased expression of corn *ZMPTF1* regulates its drought resistance [38]. In *Arabidopsis*, *AtbHLH112* is a transcriptional activator that regulates the expression of genes by binding to their GCG- or E-boxes to mediate physiological responses, including proline biosynthesis and ROS scavenging pathways, to enhance stress tolerance [39]. These results indicate that identifying key drought-resistant bHLH TFs is of great value for drought resistance molecular breeding in plants. In this study, PEG treatments were used as relevant tests of the potential for plant drought tolerance, and a total of 128 *bHLH* genes were identified in birch genome and important drought-resistance-related *BpbHLH* genes were screened and identified through sequence characteristic analysis, phylogenetic analysis, tissue-specific expression analysis, and expression analysis of different tissues under osmotic stress. These provide genetic resources for studying the molecular mechanism of drought resistance and drought resistance breeding in *B. platyphylla*.

Conserve motif analysis and promoter cis-element investigation indicate that the promoters of *BpbHLH* genes contain a large number of cis-elements, such as MYB- and MYC-binding sites (Figure 3B), which previous studies have shown respond to MeJA, ABA, SA, and GA [40,41]. JAM1 (ABA-inducible bHLH-type TF/JA-associated MYC2-like1) negatively regulates JA signaling, thereby playing a pivotal role in the fine-tuning of JA-mediated stress responses and plant growth. Under drought stress, *Arabidopsis AtMYC2* (*bHLH*) functions as a transcriptional activator in abscisic acid signaling and AtMYC2 and AtMYB2 proteins function as transcriptional activators in ABA-inducible gene expression in plants [42]. A number of cis-acting elements related to plant hormones and stress responses were also predicted in the promoter regions of *SabHLHs*, and most were involved in tolerance to drought and salinity [43]. In this study, the expression profile based on RNA-seq showed that most of the *BpbHLH* genes responded to osmotic treatment, indicating that these genes might be involved in drought resistance regulation mediated by the hormone signaling pathway in birch.

According to the NJ phylogenetic tree, the *BpbHLH* members can be clustered into 21 different subfamilies (Figure 2). Motif analysis showed that each group has a specific motif (Figure 3), which corresponds to a specific protein domain. In addition, each group has similar intron and exon structures (Figure 2). The structure and motif analysis results of *BpbHLHs* are similar to those of *Arabidopsis*. In phylogenetic tree analysis, some *BpbHLHs* have been clustered into one group with *AtbHLHs*, which play important roles in response to drought stress. The *Arabidopsis BEE* gene is homologous with *BPChr12G24135* (Group III, Appendix A). *BEE* is a classic bHLH protein. The *bee1/bee2/bee3* mutant shows enhanced drought tolerance, and the dual mutant behavior is similar to that of wild plants [44]. *BPChr12G24135* is the gene with the lowest relative expression in roots, stems, and leaves of birch (Figure 5), but it is significantly upregulated under osmotic stress conditions (Figure 6). These results suggest the potential roles of *BPChr12G24135* in plant response to osmotic stress. *HEC1* and *HEC2* are two TFs in the bHLH family and are homologous to *BPChr12G25787* (Group IV). The excessive relative expression of *HEC1* and *HEC2* stabilizes the *HEC2* protein at high temperatures, promoting the resistance of *Arabidopsis* to drought [45]. In this study, using qRT-PCR to prove *BPChr12G25787* is one of the genes that could be induced to high levels of expression under osmotic treatment in roots, stems, and leaves (Figure 5). What is interesting is that among the eight *BpbHLH* genes analyzed by qRT-PCR, *BPChr12G25787* was relatively less expressed in roots, stems, and leaves, implying that the relatively less expressed genes in tissues also function in stress response in plants. The results (Figure 5 and Figure 6) confirmed significant induction of these eight *bHLH* genes under osmotic stress, with varying expression patterns over time. Notably, the expression levels of some of these genes reached a peak at specific time points. Production discontinuity or delay is an inevitable part of gene expression. It can be caused by a number of mechanisms, e.g., transcriptional/translational elongation, post-translational modification, or compartmental transport. The delay specifies the amount of time that needs to pass before a newly produced molecule can partake in its regulatory function (specifically in feedback) [46].

In addition to studies on *bHLH* family genes in *Arabidopsis*, there are also many studies on the role of *bHLH* genes in drought stress in other plant species. *AhHLH112*, a bHLH TF in peanut, is homologous to *BPChr11G07048* (Appendix A). Studies have shown that *AhbHLH112* improves ROS scavenging ability by regulating the H_2_O_2_ steady state mediated by POD and may be involved in the ABA-dependent stress response pathway, which is a positive factor for drought stress tolerance [47]. *SlbHLH96*, a bHLH TF in tomato, is homologous to *BPChr12G25787* and *BPChr11G07048* (Appendix A). *SlbHLH96* improves the drought resistance of tomatoes by stimulating the expression of genes encoding antioxidant enzymes, ABA signal molecules, and stress-related proteins [48]. bHLH TFs from trees also play a positive role in drought resistance. The overexpression of the *PebHLH35* gene from *Populus euphratica* in *Arabidopsis* results in a decrease in stomatal density, stomatal aperture, transpiration rate, and water loss, as well as an increase in chlorophyll content and photosynthetic rate [49]. Both *BPChr11G07048* and *BPChr12G25787* belong to Group IV, and *BPChr11G07048* is one of the first two genes with relatively high expression levels in roots, stems, and leaves. It is also one of the most significantly upregulated genes in stems and leaves under osmotic stress treatment, indicating that the expression of *BpbHLH* genes in the same group may have similar stress response patterns, regardless of their original tissue expression levels. Therefore, the eight *bHLH* family genes play roles in the response of birch to drought.

## 4. Materials and Methods

### 4.1. Identification of bHLH Family Genes in Birch

In this study, the *bHLH* family genome sequence and all the protein sequences were obtained from the phytozome V13 database (https://phytozome-next.jgi.doe.gov/ 1 December 2020), and the HMM of the typical protein structure was downloaded from the birch genome database (https://github.com/jinzitian/HMM 2 December 2020). Hmmsearch3.0 was used to screen all potential *bHLH* family genes in birch. Then, the SMART V6.0 database was used to confirm again. The ExPASy website was used to calculate the physicochemical parameters of BpbHLH family proteins in Betula platyphylla, SignalP V4.1 Server was used to predict the cleavage site of the signal peptide, and ProtComp V9.0 was used to predict the subcellular location of BpbHLH proteins (Appendix A).

### 4.2. Intron and Exon Structures and Arrangement of Conserved Motifs

On the basis of the alignment of the coding sequence of the *BpbHLH* gene with its corresponding genomic DNA sequence, the online program GSDS (http://gsds.cbi.pku.edu.cn/ 3 December 2020) was used to display the exon and intron structures of each *BpbHLH* gene. The online tool MEME (https://meme-suite.org/meme/ 13 March 2021) was used to identify the arrangement of conserved motifs for *BpbHLH* genes. Bioedit 7.2 software was used to build the phylogenetic trees.

All upstream 2010 bp promoter sequences of *BpbHLH* family genes were obtained from the birch genome database. The online tool Place (http://bioInformatics.psb.ugent.be/webtools/plantcare/html/ 13 March 2021) was used to predict the cis-elements in *BpbHLH* gene promoters. TBTools 1.045 software was used for visual analysis.

### 4.3. Chromosomal Location and Collinearity Analysis

The identified *bHLH* family genes were mapped on the corresponding chromosome based on the birch genome data using MapChart V2.2 software. The multilinear scanning tool package (MCSCANX) was used to analyze the replication mode of each *BpbHLH* gene on the basis of the operation manual. The Circos tools in TBTools 1.045 software were used to visualize the results.

### 4.4. Gene Expression Profiles of the BpbHLH Family

A set of differentially expressed genes of birch in response to osmotic stress was identified from transcriptome analysis [34]. Three-month-old birch seedlings were treated with 20% (*w*/*v*) PEG_6000_ for 0, 2, 4, 6, and 9 h, and five groups of differentially expressed gene data were obtained through RNA-seq. Three seedlings were pooled in each sample, and three biology repetitions were carried out for each sample. The expression levels of the *BpbHLH* gene were normalized by the log_2_ ratio. Differentially expressed genes with |log_2_ Foldchange| > 1 (the single factor comparison of Prism9 software is used to define the *p*-value. *p* < 0.05) were defined as significant induction or inhibition genes, and heatmaps were generated using TBTools 1.045 software.

### 4.5. Plant Materials and Stress Treatment

Tissue-cultured birch seedlings differentiated from shoot buds were grown in 1/2 MS medium in a 16 h/8 h light/dark environment at 25 °C under 70–75% relative humidity. Three-month-old seedlings were treated with 20% PEG_6000_ for 0, 6, 12, 24, 48, and 72 h, and the non-treated seedlings at the same time point were set as the controls. The birch roots, stems, and leaves were picked at different time points. Five seedlings were pooled in each sample, and three biology repetitions were carried out for each sample. All samples were frozen with liquid nitrogen immediately and stored at −80 °C for expression analysis. 

### 4.6. RNA Isolation and Real-Time Polymerase Chain Reaction (qRT-PCR) 

The total RNA was isolated using the RNA extraction kit (Omega Bio U.S.A.), and the first-strand cDNA was synthesized using the reverse transcription kit (Omega Bio, Pinnacle Way, Norcross, GA 30071, USA). The cDNA diluted with ultrapure water was used as the qRT-PCR template. The tubulin and ubiquitin genes served as the internal controls. The primers were designed as *BpbHLH* gene sequences using Primer 5.0 (Appendix A). The qRT-PCR system contained 0.5 μM of forward and reverse primers, 10 μL of SYBR Premix Ex Taq™ (Takara, Beijing, China), and 2 μL of cDNA template, with a total volume of 20 μL. qRT-PCR was conducted on a qTower 2.2 (Analytik AG, Jena, Germany) with the following thermal profile: 30 s at 94 °C; followed by 40 cycles of 15 s at 94 °C, 30 s at 58 °C, and 45 s at 72 °C. The 2^−ΔΔCT^ was used to calculate relative transcriptional abundance. Three biological replications were performed.

### 4.7. Statistical Analysis

One-way analysis of variance (ANOVA) was employed for data analysis. The Statistical Package for the Social Sciences (SPSS 22, IBM Corp., Armonk, NY, USA) was used to perform all the statistical analyses, and *p* < 0.05 was considered to indicate statistical significance.

## 5. Conclusions

In all, 128 *bHLH* genes were identified in the birch genome and were randomly distributed to all 14 birch chromosomes. The distribution patterns of motifs and exons were relatively conserved in the group. The promoters of *BpbHLH* genes contain a large number of hormone-, light-, and stress-responsive elements. The *BpbHLH* gene response to PEG osmotic stress had tissue-specific characteristics. The eight *bHLH* genes selected played a crucial role in the osmotic stress resistance process of birch leaves.

## Figures and Tables

**Figure 1 plants-12-03687-f001:**
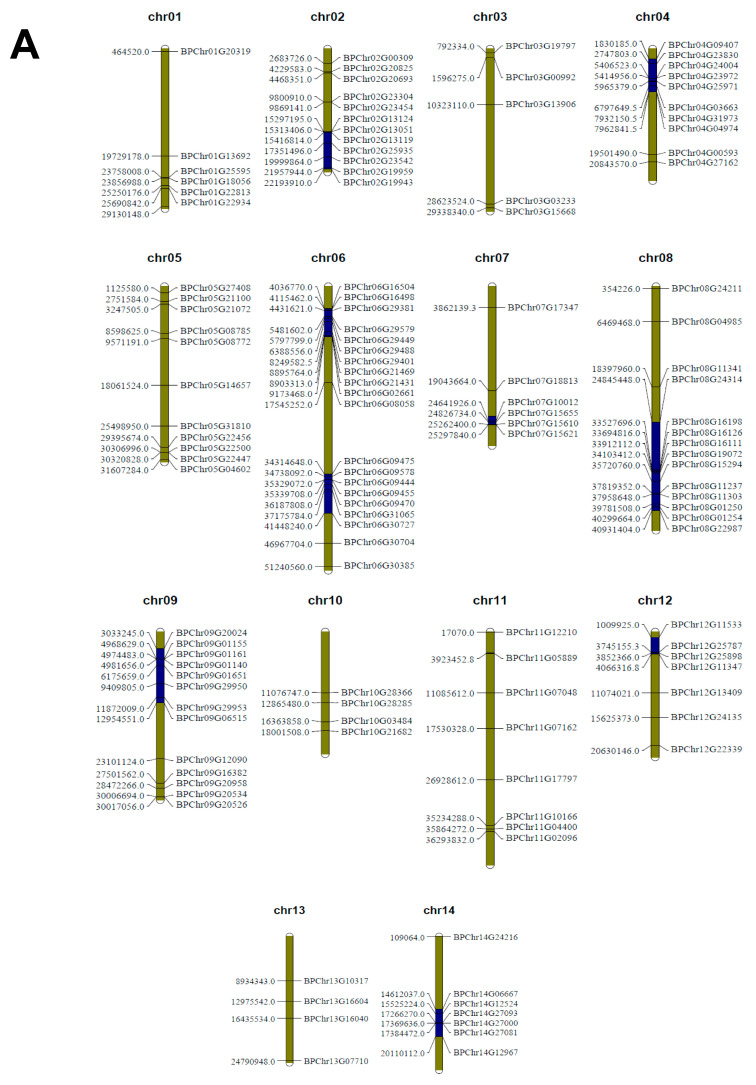
Chromosomal localization and collinearity analysis of *BpbHLH* family genes. (**A**) Distribution of *BpbHLH* family genes in the 14 birch chromosomes. The vertical strips represent the birch chromosomes. The chromosome number is located at the top of each chromosome. The left side of the chromosome represents the position of the gene on the chromosome, and the right side of the chromosome is the gene ID. (**B**) Collinearity analysis of *bHLH* genes in birch. The circle plot represents the chromosomes of birch, and the identified collinear genes are linked by blue lines.

**Figure 2 plants-12-03687-f002:**
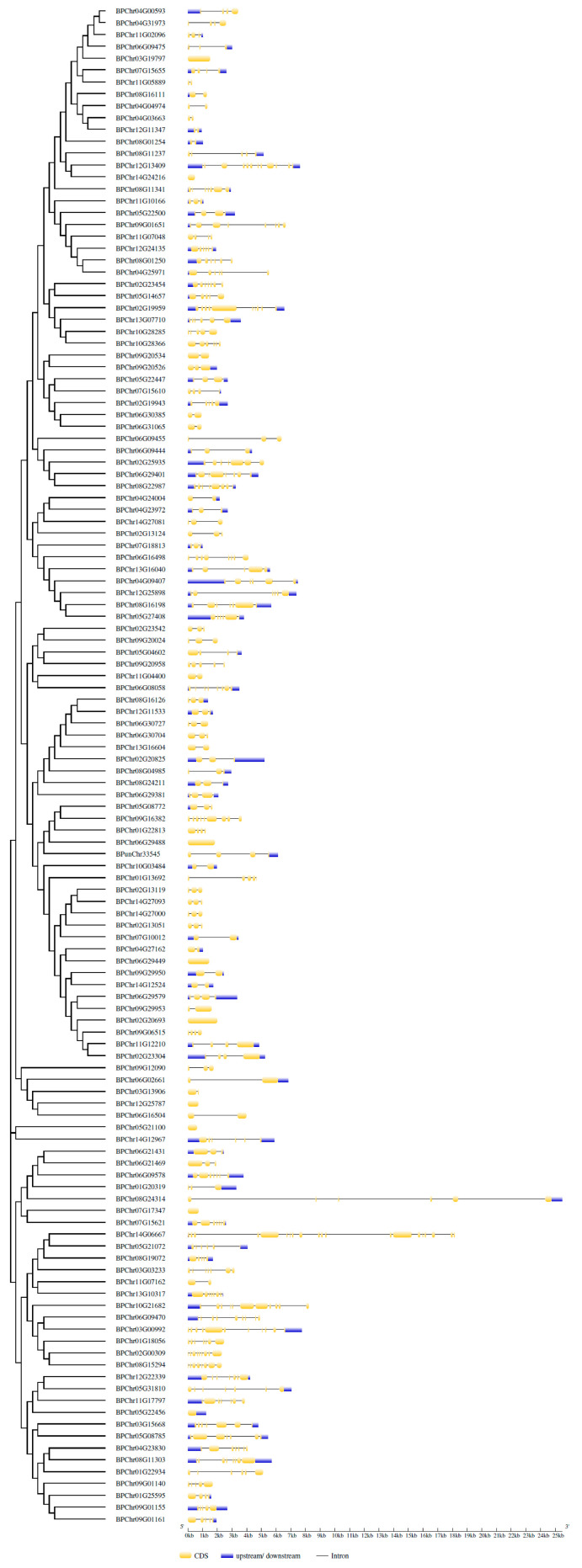
Gene structure of BpbHLH family proteins. For gene structure analysis, exons and introns are shown in yellow boxes and black solid lines, respectively. Cluster analysis was carried out according to the results of phylogeny.

**Figure 3 plants-12-03687-f003:**
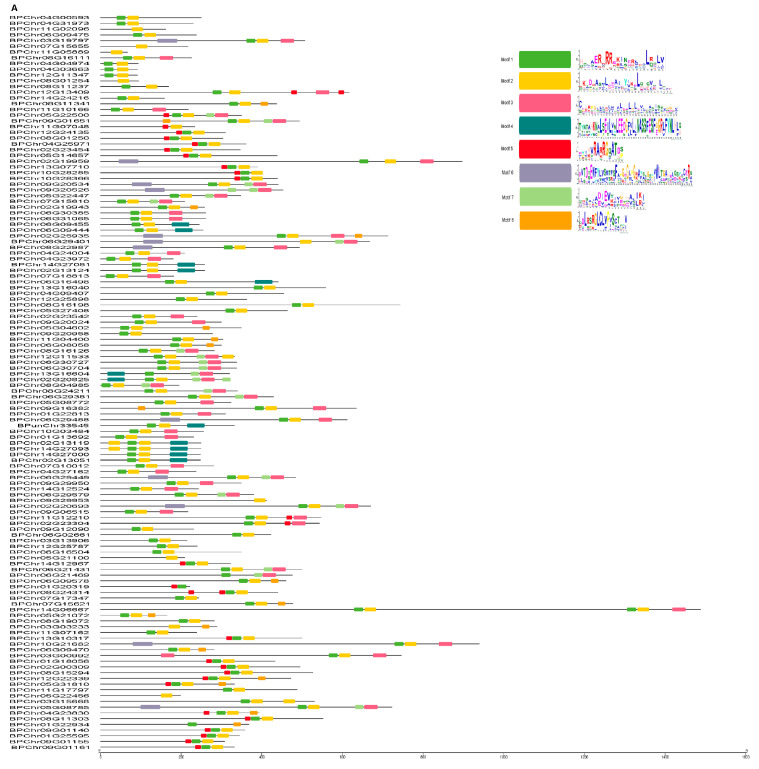
Protein motif and analysis of the cis-regulatory element in the promoter of *BpbHLH* family genes. (**A**) Analysis of protein motifs. Different motifs are displayed in boxes of different colors. (**B**) Prediction of cis-elements in the *BpbHLH* gene promoter. On the left is the gene ID, and different patterns on the right represent different cis-elements.

**Figure 4 plants-12-03687-f004:**
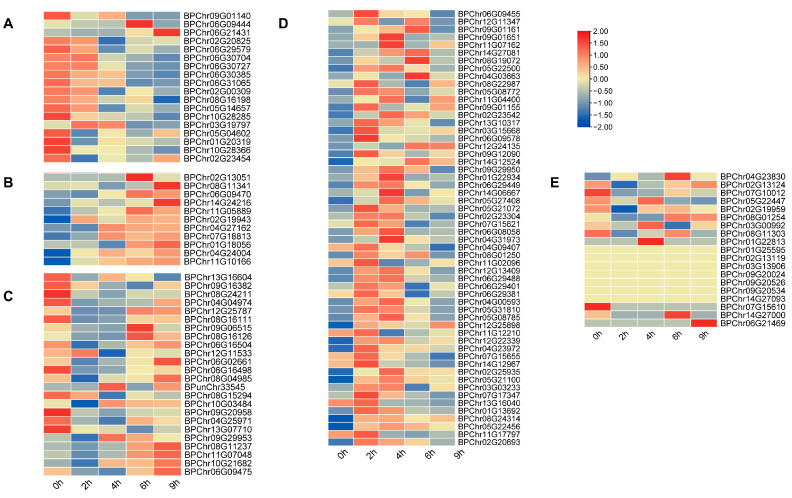
Expression profiles and gene replication events of BpbHLH family genes. The heatmap reflects the expression changes in BpbHLH genes in response to osmotic stress of whole plant for 0, 2, 4, 6, and 9 h. Red represents upregulation, and green represents downregulation. Class (**A**) represents genes that are downregulated under osmotic drought, Class (**B**) represents genes that are upregulated under osmotic drought, and Class (**C**) represents genes that are downregulated in early BHLH gene expression and upregulated in late BHLH gene expression during osmotic drought; Class (**D**) represents genes that upregulate the expression of BHLH genes in the early stage and downregulate BHLH genes in the late stage of osmotic drought, while Class (**E**) represents genes that do not show significant changes or responses to osmotic drought. The expression values were converted by log2. The error bar indicates the standard deviation (STD) of the three biological replicates (*p* < 0.05).

**Figure 5 plants-12-03687-f005:**
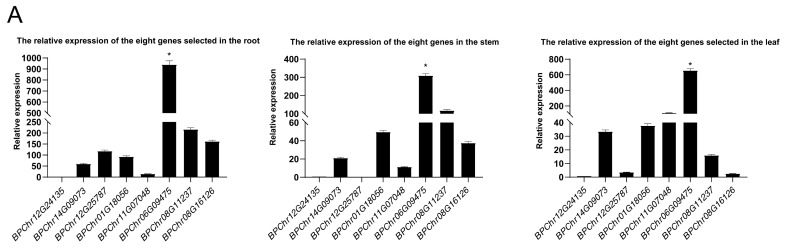
Expression of eight selected *bHLH* genes in different tissues of birch. Tissues from 3-month-old plants were used for the analysis. The relative expression of *BPChr12G24135* in different tissues of birch was set as 1 to normalize its expression in other tissues. (**A**) The relative expression of eight selected *bHLH* genes in the roots, stems, and leaves of birch. (**B**) The relative expression level of eight different genes in the roots was set to 1 to normalize the expression of other genes in the stems and leaves. The relative expression levels of the eight selected genes in the roots, stems, and leaves. The expression value is 2^−ΔΔt^. The error bar indicates the standard deviation (STD) of the three biological replicates. The asterisk (*) represents significant difference compared with *BPChr12G24135* (*p* < 0.05).

**Figure 6 plants-12-03687-f006:**
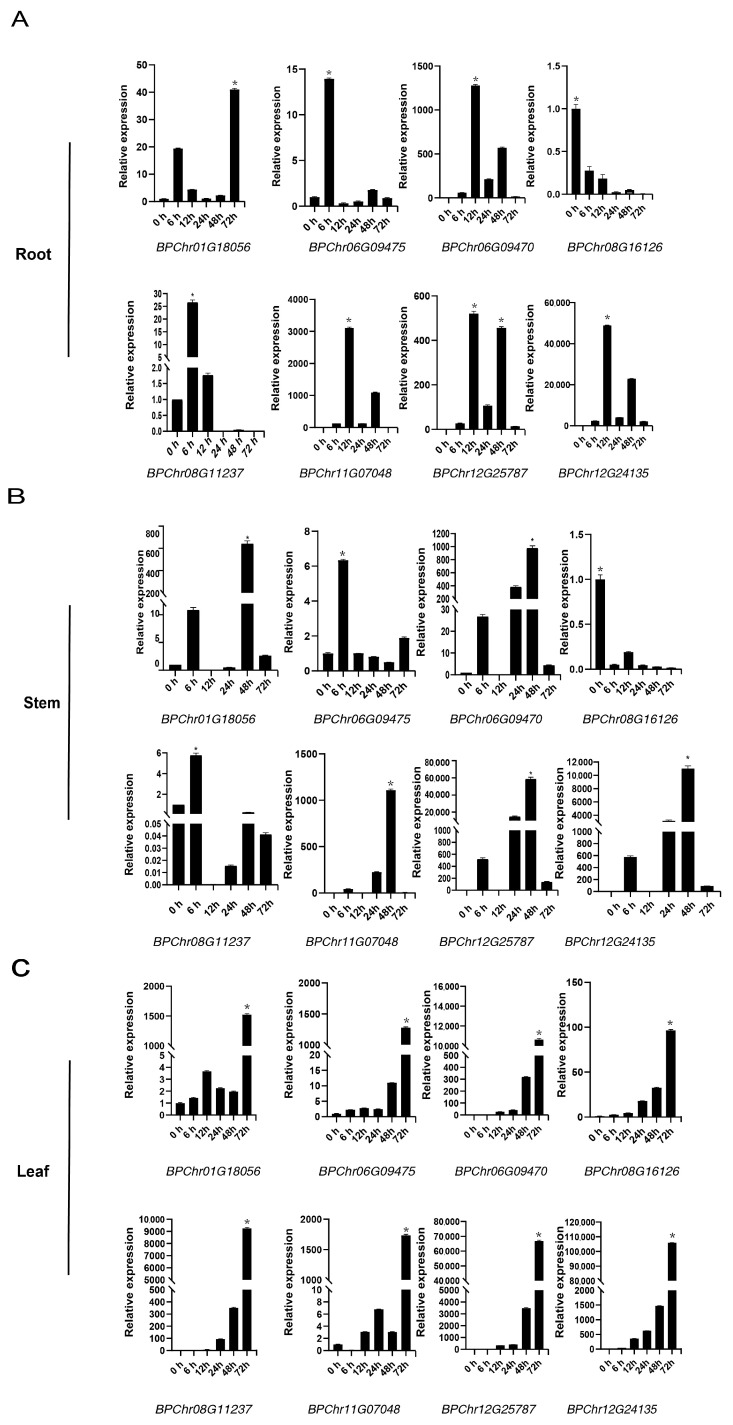
Three-month-old Betula platyphylla was treated with 20% PEG_6000_ for 0, 12, 24, 48, and 72 h. The relative expression of eight *bHLH* genes in roots, stems, and leaves was obtained. (**A**) Relative expression of the eight *bHLH* genes in the roots of *B. platyphylla* at different time points. (**B**) Relative expression of the eight *bHLH* genes in the stems of *B. platyphylla* at different time points. (**C**) Relative expression of the eight *bHLH* genes in the leaves of *B. platyphylla* at different time points. The expression value is 2^−ΔΔt^. The error bar indicates the standard deviation (STD) of the three biological replicates. The asterisk (*) represents significant difference compared with 0 h (*p* < 0.05).

## Data Availability

The original contributions presented in the study are included in the article/Appendix A.

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
