# Peer review of "Genome-Wide Characterization of bHLH Family Genes and Expression Analysis in Response to Osmotic Stress in Betula platyphylla"

_plants, 2023, doi:10.3390/plants12213687_

Round 1
Reviewer 1 Report
The article entitled, “Genome-wide characterization of bHLH family genes and expression analysis in response to drought stress in Betula platyphylla” submitted by Dr. Wang chao, is poorly written and represented by figures and it need improvement before publishing,.
Suggested minor revision-
- There are many grammatical and spelling errors throughout the manuscript which needs to be corrected after thorough reading.
- Authors need to improve language as in many sentences in the manuscript as its fail to explain its meaning.
- Few lines mentioned below are example which should be also checked and corrected after thorough revision-
Line no. 9: spelling of “analysis” as “ananlysis”.
Line no. 15-16: Discussion about cis elements requires rephrasing.
Line no. 65: unnecessary addition of design word.
Line no. 73: “In this study the bHLH family members of were identified and the”-mistakes like this throughout the manuscript.
Line no. 75: Rephrasing required.
Line no. 145: “et” or “etc”?
Line no. 297: Spelling “Homologe”?
Line no. 335: Spelling “Arrengement”?
- Also,
1- Introduction is short and basic. It needs improvement and addition of more examples as function of bHLH is very well characterized in model plant like Arabidopsis.
2- Discussion of its regulatory mechanism in introduction is required.
3- The picture quality is dull and fails to provide satisfactory explanation of any result.
4- Marking in picture is not clear, figure legend is not written properly.
5- The relative expression of gene with tissue type cannot be the true representation of positive correlation of that gene in any stress or developmental stage. It always needs further validation with other genes which has been proven to have involvement in that particular stress.
6- Proper citations should be added, manuscript is lagging in citations.
Corrections required
Author Response
Dear Reviewer,
Thank you very much for taking the time to review this manuscript. The comments you have made have been correspondingly modified and corrected in the submitted document. Please see the attachment.

Reviewer 2 Report
The study presents interesting results. The study is quite well introduced, providing a good background. I consider the data as worth to be published, but Major revision is required, after which the manuscript could become acceptable for publication.
I have few comments:
“Differential expression levels of BpbHLH genes un-16 der drought stress was determined using RNA-seq data, including 8 highly differentially expressed 17 BpbHLH genes. The qRT-PCR analysis results further indicated that BpbHLH056 was the gene with 18 the highest expression level in leaves roots and stems, and expression of all of these eight genes 19 were higher in leaves than those in roots and stems and show a positive correlation with drought 20 stress in all three tissues”. In this lines authors said that its positively correlated with drought. How?
Also I would suggest authors to examine some downstream genes which correlated with drought tolerance. Basically interaction of this transcription factor with other genes.
I will suggest add 2 or 3 more keywords.
Introduction part need to be improved in terms of importance of transcription factor, plant material uses etc. also English need to be improve.
Throughout the manuscript figure quality is not good. Please improve figures with high resolution.
Throughout the manuscript Arabidopsis written somewhere italic and somewhere normal . Please check and do it uniform.
Must be improve
Author Response

(The authors gave the same response as above.)

Round 2
Reviewer 1 Report
I could see the Ms has been thoroughly revised, but it seems to be still not improved scientifically and even technically. For example, gene name and protein name.
Extensive editing required
Author Response
Dear reviewer
Thank you for your comments on the second revision of our manuscript. We apologize for wasting your precious time due to some problems with our manuscript.
Thank you again for taking the time to review our manuscript. We apologize for the problems with our manuscript.
In response to your comments on the manuscript, we have carefully revised it. For example, the question of whether bHLH is in block letter or italic.

Reviewer 2 Report
Figure quality is still not good
Author should improve the figures
Author Response
Dear Reviewer
Thank you for your comment on the second revision of our manuscript.
Regarding your comment on our manuscript, the image is unclear. We further optimized the images in the manuscript. For example: Figure 1, Figure 2, and Figure 3.
We deeply apologize for any unpleasant experience.
Round 3
Reviewer 1 Report
NA
Needs correction.
Author Response
Dear Reviewer
Thank you for your feedback.
After the first round of the review, we had undergone English language editing by MDPI. If the reviewer believes that further editing will be required, please allow us to extend the time for another revising by MDPI.
And we have also made further modifications to the manuscript, such as language, grammar, and spelling.
Looking forward to your reply.
Thanks again!
Best.
